# Natural Products with Antitumor Potential Targeting the MYB-C/EBPβ-p300 Transcription Module

**DOI:** 10.3390/molecules27072077

**Published:** 2022-03-23

**Authors:** Thomas J. Schmidt, Karl-Heinz Klempnauer

**Affiliations:** 1Institute of Pharmaceutical Biology and Phytochemistry (IPBP), University of Münster, PharmaCampus-Corrensstraße 48, D-48149 Munster, Germany; 2Institute of Biochemistry, University of Münster, Corrensstraße 36, D-48149 Munster, Germany

**Keywords:** transcription factor, MYB, C/EBPβ, p300, cancer, leukemia, natural product, sesquiterpene lactone, withanolide, withaferin A, celastrol, naphthoquinone, plumbagin

## Abstract

The transcription factor MYB is expressed predominantly in hematopoietic progenitor cells, where it plays an essential role in the development of most lineages of the hematopoietic system. In the myeloid lineage, MYB is known to cooperate with members of the CCAAT box/enhancer binding protein (C/EBP) family of transcription factors. MYB and C/EBPs interact with the co-activator p300 or its paralog CREB-binding protein (CBP), to form a transcriptional module involved in myeloid-specific gene expression. Recent work has demonstrated that MYB is involved in the development of human leukemia, especially in acute T-cell leukemia (T-ALL) and acute myeloid leukemia (AML). Chemical entities that inhibit the transcriptional activity of the MYB-C/EBPβ-p300 transcription module may therefore be of use as potential anti-tumour drugs. In searching for small molecule inhibitors, studies from our group over the last 10 years have identified natural products belonging to different structural classes, including various sesquiterpene lactones, a steroid lactone, quinone methide triterpenes and naphthoquinones that interfere with the activity of this transcriptional module in different ways. This review gives a comprehensive overview on the various classes of inhibitors and the inhibitory mechanisms by which they affect the MYB-C/EBPβ-p300 transcriptional module as a potential anti-tumor target. We also focus on the current knowledge on structure-activity relationships underlying these biological effects and on the potential of these compounds for further development.

## 1. The MYB-C/EBPβ-p300 Transcription Machinery: Its Physiological and Pathophysiological Roles and Its Potential as a Target in Cancer

*MYB.* The *MYB* proto-oncogene is the founding member of the *MYB* gene family that consists of *MYB*, *MYBL1* (A-*MYB*) and *MYBL2* (B-*MYB*) in vertebrate species (for review see: [1,2]). *MYB* genes encode transcription factors with a similar domain structure, consisting of an N-terminal DNA-binding domain, a centrally located transactivation domain (TAD), and C-terminal regulatory sequences. All *MYB* family members are involved in important aspects of cell proliferation and differentiation. In vertebrates, *MYB* is predominantly expressed in the hematopoietic progenitor cells. A large body of evidence, derived from the study of *MYB* null and conditional knock-out mice and the analysis of hypomorphic *MYB* alleles, has demonstrated an essential role for *MYB* in the development of most lineages of the hematopoietic system [3,4,5,6,7,8]. *MYB* is also expressed in certain non-hematopoietic tissues [9] during embryonic development, and has been implicated in the proliferation of the intestinal stem cells in the colonic crypts in the adult [10].

The first gene to be identified as a direct *MYB*-regulated target was the chicken *MIM*-1 gene (*MIM*-1 stands for “MYB-inducible myeloid-specific gene 1”) [11]. *MIM*-1 expression is stimulated by MYB (the protein encoded by the *MYB* gene), specifically in cells of the myelomonocytic lineage, and reaches very high levels due to the binding of MYB to several high-affinity MYB-specific DNA-binding sites located in the promoter region, as well as in a myeloid-specific enhancer located upstream of the *MIM*-1 promoter [12]. Analysis of the cell-type specificity of *MIM*-1 expression has identified members of the CCAAT-box enhancer protein (C/EBP) family as crucial MYB cooperation partners [12,13,14,15]. Notably, MYB and a C/EBP family member, such as C/EBPα or C/EBPβ, were sufficient to induce *MIM*-1 expression in cell types in which the gene is normally silent. Analysis of the chromatin structure of the *MIM*-1 gene has shown that MYB and C/EBPβ cooperate in opening the compact chromatin structure of the gene at its regulatory sequences in *MIM*-1 non-expressing fibroblasts, and has revealed a specific role of C/EBPβ in triggering the initial step of chromatin opening [16]. Subsequent studies of MYB’s role in gene expression have revealed numerous MYB-regulated genes with relevant roles in cell proliferation, survival and differentiation [17,18,19].

MYB’s role as a transcription factor is highly dependent on the cooperation with the co-activator p300 or its paralog CREB-binding protein (CBP) [20,21]. p300/CBP is involved in direct protein-protein-interactions with MYB, which are mediated by the KIX domain (kinase-inducible domain (KID) interacting domain) of the co-activator and a LXXLL amino acid motif present in the TAD of MYB [22] (Figure 1). The relevance of this interaction has been established by point mutations of the LXXLL motif, such as M303V or L302A, or of the KIX domain of p300, which disrupts the MYB-p300 interaction and results in the decreased or virtual loss of the transactivation potential of MYB. Mice that harbor such mutations show severe defects of their hematopoietic system, further demonstrating the relevance of MYB and its interaction with p300 for the development of the hematopoietic system [7,23,24,25]. The *MYB* gene was originally identified as the cellular progenitor of the leukemia oncogene v-*myb* of the avian myeloblastosis virus, an agent that induces myeloid leukemia in chickens [26]. It was therefore suspected that *MYB* may also be involved in the development of leukemia in humans. This conjecture has proven correct, especially in the case of acute T-cell leukemia (T-ALL) and acute myeloid leukemia (AML). In T-ALL, chromosomal rearrangements affecting the *MYB* gene have repeatedly been identified [27,28,29]. In addition, recent work has shown that point mutations can generate de novo MYB binding sites in transcriptional control regions that affect the expression of other genes, as shown for the *TAL1* or *LMO2* oncogenes [30,31]. In AML, MYB rearrangements are rare; however, AML cells are often addicted to higher levels of MYB expression than normal hematopoietic progenitor cells, making the leukemia cells more vulnerable to MYB inhibition than their normal counterparts [32,33,34]. Besides leukemia, overexpression of MYB has also been implicated in the development of other malignancies, such as breast, colon and prostate cancer, adenoid cystic carcinoma (ACC) and low-grade glioma [35,36,37]. Recurrent translocations fusing *MYB* with the *NFIB* gene and leading to the expression of oncogenic MYB-NFIB fusion proteins are detected in a high percentage of ACC cases [38]. Translocations fusing *MYB* with different other loci occur frequently in low-grade pediatric gliomas [39,40]. Importantly, the DNA-binding and transactivation functions of MYB are usually retained in the resulting fusion proteins; however, they are expressed at elevated levels as a result of the translocation. For example, in the case of *MYB-NFIB* translocations in ACC, the translocation positions a MYB-dependent super-enhancer towards the *MYB* gene, thereby establishing a positive feedback loop, which leads to elevated *MYB* expression [41].

*C/EBP.* CCAAT box/enhancer binding proteins (C/EBPs) form a family of basic-region-leucine-zipper (bzip) transcription factors that are involved in the control of differentiation and proliferation of various cell types, including adipocytes, keratinocytes, mammary epithelial cells and myeloid cells [42,43,44]. In the myeloid lineage of the hematopoietic system, several C/EBP family members control the transcription of genes specifically expressed during the differentiation of the cells. C/EBPs are subject to a variety of post-transcriptional and post-translational mechanisms that modulate their activity, as exemplified by C/EBPβ, a well-studied C/EBP family member. On the translational level, C/EBPβ is subject to alternative translational initiation of its mRNA, thereby generating three isoforms with different N-termini: A full-length isoform referred to as LAP* (LAP stands for the alternative name “liver activating protein”), and two shorter isoforms, lacking 21 (LAP) or 110 (LIP, for “liver inhibitory protein”) N-terminal amino acids as a result of translation initiation at internal in-frame AUG codons [45,46]. Since the transactivation domain of C/EBPβ is encoded by N-terminal amino acid sequences, LAP* and LAP are transcriptional activators whereas LIP lacks transactivation potential. The activity of C/EBPβ is additionally regulated by post-translational modifications, including phosphorylation, acetylation, methylation and sumoylation, which orchestrate its function by modulating interactions with other proteins [47,48,49,50,51,52,53,54]. Like MYB, C/EBPβ also interacts with the co-activator p300 [55]. However, both proteins employ different domains of p300 as docking sites. While MYB binds to the KIX domain of p300, C/EBPβ interacts with the TAZ2 (transcriptional adaptor zinc finger 2) domain of the co-activator (Figure 1). In addition to p300, C/EBPβ interacts with the ATP-dependent chromatin-remodelling factor SWI/SNF and the mediator complex of the basal transcriptional machinery [54,56]. Most of these interactions are mediated by conserved amino acid sequence motifs that are embedded in the largely unstructured N-terminal TAD. Recent proteome-wide binding studies have revealed a multitude of further proteins interacting with C/EBPβ, thereby underlining the complexity of its cellular functions [57].

Deregulation of C/EBPβ expression has been implicated in the development of certain malignancies. Increased expression of C/EBPβ is found in colon, kidney, stomach, prostate and ovarian tumors, and to correlate with increased malignancy and invasiveness of the tumor cells in many cases [58,59,60]. In glioblastoma, high C/EBPβ expression predicts a poor prognosis for patients [61,62]. In these cells, C/EBPβ has been implicated in establishing a mesenchymal gene expression signature that is responsible for the aggressiveness of high-grade glioblastomas [63]. In the hematopoietic system, C/EBPβ is a pro-oncogenic factor in ALK-positive anaplastic large cell lymphoma (ALCL) [64,65] as a crucial downstream target of oncogenic ALK-signalling. Finally, as described in more detail below, our own recent work has implicated C/EBPβ as a pro-oncogenic factor in AML [66,67].

Taken together, these lines of evidence make it likely that inhibitors of the MYB and C/EBPβ transcription factors would have an interesting potential as leads or drugs against leukemia and, possibly, other MYB- or C/EBPβ-dependent tumors.

## 2. Development of a Fluorescence-Based MYB Reporter Cell-Line

As increasing evidence has implicated MYB in human leukemia, we became interested in developing a screening tool that would allow us to identify compounds with MYB-inhibitory activity. The extremely strong stimulation of *MIM*-1 expression by MYB and the identification of the cis-acting sequences mediating this activation [12] prompted us to construct a reporter cell line based on these highly MYB-responsive elements (Figure 2). Thus, we combined the *MIM*-1 promoter and enhancer with the coding sequence of the green fluorescent protein (GFP), and integrated them stably into the genome of the myeloid chicken cell line HD11, which does not express MYB [68]. Additionally, we equipped these cells with a doxycycline-inducible expression system for MYB. The resulting cells showed a significant increase of their fluorescence when MYB expression was induced by doxycycline. We then used these cells as a screening tool to identify compounds as candidate MYB inhibitors, since a decrease in fluorescence intensity in these cells exposed to a potential inhibitor in comparison with the untreated cells can be measured in a straightforward manner and is correlated with the inhibitor’s potency. In addition, doxycycline-treatment of the cells also induces the expression of the endogenous *MIM*-1 gene, providing an additional readout of the inhibitor’s potency on the level of an endogenous target gene [68]. In the course of our work, we realized that the reporter cells not only allowed us to identify MYB inhibitory compounds, but that they are also responsive to compounds that inhibit C/EBP transcription factors. This is because the reporter construct driving GFP expression was based on the natural cis-acting regions of the *MIM*-1 gene and that we used a myeloid cell line expressing endogenous C/EBP factors, particularly C/EBPβ. Thus, the stimulation of the reporter construct by MYB was also dependent on the activity of the endogenous C/EBPβ. Even though the observed fluorescence emitted by these cells may thus be due to an inhibition of the transcriptional activity of MYB or C/EBPβ or both, we will refer to this assay as “MYB assay” or “reporter gene assay” for simplicity.

## 3. Natural Products as Inhibitors of MYB-Related Transcription in the Fluorescence-Based Reporter Cell Line

Natural products (NPs) have played and still play a pivotal role in the development of drugs against cancer. The classical anticancer drugs interfering with tubulin, i.e., the Vinca alkaloids, paclitaxel and epothilones, or NP derivatives interfering with topoisomerases, such as etoposide, teniposide and the camptothecin derivatives, are just a few examples of the extraordinary importance of NPs in the development of new drugs against cancer. It is quite noteworthy that NPs have proven to be particularly important in the development of anticancer drugs over the last 40 years and are likely to continue to have substantial impact [69].

Numerous reports and comprehensive reviews exist in the literature on the potential anti-cancer activity of sesquiterpene lactones (STLs) [70,71,72,73,74], a large subclass of C-15 terpenoids from higher plants. A small selection of STLs were therefore among the first compounds to be tested in the mentioned reporter gene assay and, quite excitingly, some of the compounds, namely, helenalin and mexicanin I, showed promising activity with half-maximal inhibitory concentrations (IC_50_ values) between 1 and 3 µM [68]. Inspired by this initial finding, we then tested a library of 100 NPs of various biosynthetic classes, including many STLs, in the reporter gene assay [75]. Simultaneously, controls were carried out for unspecific cytotoxicity using an MTS assay with the same cells. A variety of NPs were found to inhibit MYB dependent fluorescence activity at IC_50_ concentrations in the submicromolar and low micromolar range. IC_50_ values in the MTS assay were generally much higher so that in most cases the inhibitory effect on MYB related transcription appeared rather selective. Overall, 22 of the 100 tested compounds (see Table 1) displayed notable activity at IC_50_ values < 5 µM, of which 13 were sesquiterpene lactones (STLs), five were quinones (three naphthoquinones (NQs), two benzoquinones (BQs)), two quinonemethide triterpenes (QMTs), one sesquiterpene dialdehyde (SDA), and one steroid lactone (SEL). A variety of NPs showed measurable but less impressive potency (IC_50_ values between 5 and 30 µM, 37 compounds), while 41 tested compounds were only weakly or not active at all (IC_50_ > 30 µM). It must be noted here that the composition of the set of test compounds was quite strongly biased towards STLs due to the initial findings [68] and availability of such compounds from earlier studies; thus, 64 of the 100 compounds belonged to this subclass of terpenoids (see Section 4 below). The structures of the most active compounds from Table 1 (IC_50_ values < 2 µM) are shown in Figure 3.

It became very clear already by this point that all compounds in the series showing significant activity in the reporter gene assay share a common chemical feature: All of them possess at least one reactive structure element in the form of an α,β- or α,β,γ,δ-unsaturated carbonyl moiety, i.e., a Michael acceptor system. Such moieties are marked in the structures of the most active compounds shown in Figure 3 and occurred in all other compounds that displayed a measurable activity in the MYB assay. The mere presence of such structural features alone, however, was found to be insufficient for strong activity, since various compounds in the set containing such features showed low or even no activity (i.e., IC_50_ > 30 µM). However, not a single compound devoid of a Michael acceptor structure did show inhibitory activity on MYB-related transcription.

## 4. Sesquiterpene Lactones

STLs represent the largest coherent subclass of terpenoids, which is particularly widespread in the largest family of higher plants, Asteraceae [76]. Many of them are well known to display a broad array of biological effects, including many reports on cytotoxic and anti-cancer activity [70,71,72,73,74]. As mentioned above, promising inhibitory activity in the fluorescence-based reporter gene assay was discovered for a small set of STLs, helenalin and mexicanin I (MEX) being the most active in this series [68]. Because it was of interest to investigate this class of NPs more closely in terms of structure-activity relationships as well as with regard to the exact mechanism of action, we then acquired data for over 60 sesquiterpene lactones (STLs) in the mentioned screening [75,77]. The activity level of the STLs in the screening in terms of half maximal inhibition concentrations (IC_50_) ranged from 0.6 and 0.7 µM in the case of goyazensolide (GOY) and helenalin acetate (HAC), respectively, up to >30 µM. As many as 42 of the 64 STLs (65%) displayed activity with a measurable IC_50_ (i.e., ≤30 µM), while the remaining 22 compounds were below this threshold. As already mentioned above, the cytotoxicity was generally much lower. Furthermore, there was only a weak correlation between the IC_50_ values for MYB inhibition and the impairment of cell viability so that an effect due to general cytotoxicity on the assay cell line could be ruled out. In view of the considerable structural diversity of the tested STLs, it appeared feasible to conduct a quantitative structure-activity relationship study [77]. In this study, correlations between the structural properties of 60 STL and their inhibitory activity in the MYB assay were analyzed (some compounds from the initial screening such as anisatin and artemisinin were left out of the study due to atypical structures since they did not show any significant activity).

As already mentioned, with regard to the whole set of NPs, compounds lacking Michael acceptor structure elements were inactive, but the mere presence of such reactive sites alone did not warrant inhibitory activity either. Compounds of the helenalin type (helenalin and various ester derivatives represented by helenalin acetate (HAC) as most active derivative of this type) combining an α-methylene-γ-lactone and a 2,3-cyclopentene-1-one moiety were among the most active STLs and were much more potent than various tested derivatives of 2,3- and/or 11,13-dihydrohelenalin. The presence of (at least) two Michael acceptor sites might hence be concluded to be crucial for high activity. However, synthetic cyclopent-2-enone and α-methylene-γ-lactone, tested as model chemicals separately and also in a 1:1 mixture, did not show any inhibitory effect in the transcriptional assay [68]. Thus, the two reactive centres must be connected to each other, the STL skeleton most probably providing a suitable relative orientation to warrant the high inhibitory activity. It must be mentioned here that the conjugated carbonyl structures in STLs are soft electrophiles which favour soft nucleophiles such as the SH groups of cysteine (Cys) residues in proteins as reaction partners [78,79,80]. The relatively simple structure-activity relationship up to this point therefore indicated a mechanism of action involving alkylation of one or several cysteines in the target protein. Notably, the furanoheliangolide-type STL GOY, with a structure appearing, at first sight, rather dissimilar from the pseudoguaianolide-type STLs of the helenalin group, was even slightly more potent than HAC. A detailed comparison of the 3D structures of these two most potent inhibitors revealed a rather high degree of similarity in molecular shape, hydrophilic/hydrophobic surface area and electrostatic properties (Figure 4A,B). Their reactive structure elements are indeed in a similar relative orientation when their structures are aligned with respect to their main pharmacophoric properties yielding the molecular alignment shown in the middle panel of Figure 4A. A 3D pharmacophore hypothesis could thus be established from the common structural features of the two most active STLs (Figure 4C), which was then used to construct a three dimensional quantitative structure-activity relationship (3D QSAR) model. Each of the other STLs was aligned with this pharmacophoric arrangement, and the presence/absence of various structural features in their structures then expressed in terms of binary variables of type 1/0. The resulting binary pharmacophore descriptors together with some other molecular descriptors were analysed by partial least squares regression, yielding a 3D QSAR model which explains in a very satisfactory manner the variance in the measured biological data and also allows reasonably precise predictions for external compounds. The common pharmacophore underlying this model comprised various regions in space in which the Michael acceptor structures, as well as a hydrogen bond acceptor, were located (arrows in Figure 4C). The variance in activity among the 60 STLs, according to this model, is explained very well by the degree to which their structures match the various pharmacophore regions, in combination with the overall fit of their alignment with the two most active molecules and with the surface distribution of partial charge [77].

Most importantly, the QSAR model strongly supports the idea that inhibitory activity in the reporter gene assay is not only related to the presence or absence of Michael acceptor sites but that it is strongly dependent on their relative orientation in space (with respect to the STL scaffold). This points towards a highly specific interaction with the putative common binding site at the target protein.

In a series of detailed mechanistic studies on several STLs [66,81,82], it was found that the activity of these compounds observed in our MYB assay is more complex than believed at the time when the above-mentioned screening and QSAR studies were conducted. As already mentioned, MYB exerts its transcriptional activity in close cooperation with other proteins. Investigations on the effect of the pseudoguaianolide HAC [81,82], as well as several furanoheliangolides such as the 4,15-iso-Atriplicolide (AT) derivatives [66], revealed that these compounds do not directly affect MYB but actually act as inhibitors of C/EBPβ. The inhibitory effect of STLs observed in the MYB reporter gene assay was actually caused by the inhibition of the function of C/EBPβ as a co-operation partner of MYB at the promoter and enhancer elements of the *MIM*-1 gene used in the reporter construct, and not by direct inhibition of MYB activity [66,81,82].

As shown in Figure 1, C/EBPβ is thought to interact via three distinct sequence motifs within its TAD with the TAZ2 domain of co-activator p300 to stimulate the transactivation potential of C/EBPβ. HAC was shown to disrupt the interaction of the TAZ2 domain with the N-terminal part of the C/EBPβ transactivation domain, which covers the LAP*-specific N-terminal extension and adjacent amino acid sequences, resulting in a greatly reduced stimulation of the C/EBPβ transactivation potential by p300 [82]. This explains why, in the removal of the first 21 amino acids of C/EBPβ (as in case of the LAP isoform), the protein is much less sensitive to HAC than the LAP* isoform. We could show by microscale thermophoresis (MST) that HAC binds to the full-length isoform, i.e., LAP*, but not to the 21 amino acids in the shorter LAP isoform. This suggests that HAC may directly compete with the binding of p300 to this region or affect the structure of the N-terminal part of C/EBPβ in a manner that prohibits its interaction with p300. Mutation of a conserved FYY amino acid motif immediately adjacent to the LAP* specific sequences also reduced the sensitivity to HAC and prevented its binding [82]. It is therefore possible that the binding site for HAC is not entirely confined to the LAP*-specific amino acid sequences but may also involve immediately adjacent sequences from the N-terminus of LAP. However, as mentioned before, LAP cannot bind HAC because it lacks the first 21 N-terminal amino acids of LAP*. Moreover, the related C/EBPα also lacks sequences corresponding to the first 21 amino acis of C/EBPβ at its N-terminus and was found to be virtually insensitive to HAC [81] This further supports the relevance of the LAP*-specific sequences for the inhibition of C/EBPβ by HAC.

Since cysteine (Cys) residues were expected to be targeted by the reactive Michael acceptor sites of the STL, site-directed mutagenesis experiments were conducted in which two cysteines (C11 and C33) occurring in the N-terminal region of C/EBPβ were replaced by alanine. However, the sensitivity of the mutant proteins towards HAC was only decreased, but not completely abolished in comparison with the wild type, and the binding affinity of the C11A/C33A mutant towards HAC was still about one third of the wild type protein, as determined by MST. The slight reduction in binding and inhibitory potential observed in the C11A/C33A mutant might simply be due to subtle changes (e.g., steric effects induced by the difference between Cys and Ala) in the HAC-binding site in the N-terminal region of C/EBPβ. Furthermore, none of these Cys residues was found to be irreversibly modified by HAC according to mass spectrometric experiments [unpublished]. Thus, the available evidence made it likely that the binding of HAC to C/EBPβ is reversible [82] and not caused by covalent modification of Cys residues.

It was demonstrated very recently that a synthetic compound combining the reactive structure elements of HAC in a simplified molecular scaffold (“helenalin-mimic” HM, Figure 5A) is also an inhibitor of C/EBPβ, with slightly reduced potency in comparison with the native STL [83]. Although this compound is little more than a combination of cyclopentenone and α-methylene-γ-lactone, connected via a one-carbon linker mimicking carbon 6 of HAC, it was thus much more potent than a 1:1 mixture of the two isolated model compounds which had previously been found inactive (see above). Its lower activity in comparison with the native STL may be related to its more flexible molecular structure. The conformational space of helenalin derivatives such as HAC is restricted to two distinct, but energetically very similar, conformations with respect to the seven-membered ring [84] (Figure 5B left). It is clear that the molecule must adopt either one of these when binding to its target. A conformational analysis of the molecular scaffold of HM (the ester group replaced by acetate for analogy) conducted for this review resulted in 15 distinct conformers within an energy window of 5 kcal/mol above the global minimum (Figure 5B right), of which those with the lowest energy are rather different from those possible in the case of HAC (Figure 6B middle). In order to adopt one of the HAC-like conformers, i.e., a conformer in which the two main pharmacophoric elements are oriented in the same manner as in HAC, an energy expense of >3–4 kcal/mol must be invested. In other words, the fraction of molecules resembling the native STL with respect to the orientation of the two reactive structure elements in the overall conformational Boltzmann-weighted ensemble must be rather small, which could account for the reduced activity of HM. On the other hand, the native STL with its entire cycloheptane ring might be able to engage in some further hydrophobic interactions so that association with the putative target binding site might be rendered more favorable.

In contrast to HAC, HM did not show stable binding to C/EBPβ in a microscale thermophoresis experiment. However, most interestingly, but quite contrastingly, it could be demonstrated that HM does modify Cys-11 and Cys-33 in the LAP*-specific and adjoining part of the TAD, forming covalent bonds as opposed to the natural HAC. In mass spectrometric experiments after treatment of human and avian (GFP labeled) C/EBPβ proteins with the compound, both tryptic fragments containing these conserved Cys residues were found to be modified by the addition of HM [83]. The reason for this discrepancy between the synthetic mimic and the natural template is currently unclear. However, one explanation might be the fact that HM contains a third potentially reactive structure element, namely the acetylene group in the ester side chain (which was originally introduced to make it amenable to click chemistry). It is known that acetylenes do react with thiols. This may happen by means of a radical reaction (“thiol-yne-reaction” requiring a radical initiator [85,86]) but is also possible by means of a nucleophilic addition [87]. Thus, it cannot be excluded that the observed covalently modified C/EBPβ LAP* peptide fragments were actually formed by reaction with the acetylene group of HM. It will hence be highly interesting to perform experiments such as those reported in [83] with the unesterified parent alcohol of HM or with an acetate, the latter being even more similar to HAC.

As mentioned above, the furanoheliangolide-type STL goyazensolide (GOY) was as active, as HAC and several other furanoheliangolides also showed relatively strong activity in the initial MYB assay [77]. Therefore, mechanistic studies were also conducted with furanoheliangolide-type STLs [66]. Among these, 4,15-iso-Atriplicolide tiglate (AT) was also shown to inhibit C/EBPβ function by disrupting the ability to cooperate with co-activator p300. Its mechanism of action appears not to involve the alkylation of cysteine residues of C/EBPβ, as a cysteine-free mutant of C/EBPβ was inhibited equally well as the wild-type protein [66]. Given the fact that a common structure-activity relationship for so many STLs exists [77], it may, quite safely, be assumed that the general mechanism of inhibition of the function of C/EBPβ in the manner described is common to all STLs with significant activity in the MYB reporter gene assay.

The role of covalent modifications in the C/EBPβ inhibition by STLs, on the other hand, remains unclear. In view of the fact that only reactive compounds are active and that absence of reactive structure elements leads to inactivity in a series of as many as 60 different STLs, it is undoubtable that reactivity is indeed a prerequisite for activity. We repeatedly observed that the inhibitory STLs disrupt the cooperation of C/EBPβ with co-activator p300. This raises the possibility that p300 is actually the target of covalent attack by these compounds, which will have to be clarified in the future.

Even though various STLs, with helenalin among them, have been considered interesting as anti-cancer drugs or leads already in the 1970s, these compounds were later abandoned mainly because of toxicity issues (literature reviewed in [71,73,88]). However, the concentrations of HAC and GOY needed to block C/EBPβ function and thereby to inhibit transcription in the MYB assay are in the submicromolar range and we could show selectivity of this effect in comparison with an unspecific cytotoxicity [77]. Along the same lines, HAC and its synthetic analogue HM as well as the furanoheliangolide AT and its analog with a methacrylic acid ester group were proven to suppress the growth of leukemic cells taken from mice with a MLL/AF9-induced experimental acute myeloid leukaemia at concentrations that were not harmful to control hematopoietic progenitor cells from healthy mice [66,81,82]. Even more importantly, MEX [68], HAC [81] as well as AT [77] have been demonstrated to selectively suppress the growth of human acute myeloid leukaemia (AML) leukemic blasts from AML patients at concentrations (0.5, 0.5 and 0.05 µM, respectively) that did essentially not affect healthy donors’ CD34-positive myeloid progenitor cells.

Studies of the impact of all of these compounds on the gene expression profile of AML cells showed that they broadly down-modulate MYB-dependent gene expression similar to the siRNA-mediated knock-down of MYB itself. In addition, the biological effects exerted by HAC, AT, GOY and HM on AML cells are similar to those expected from the loss of MYB function, such as the induction of myeloid differentiation and cell death. Furthermore, the forced expression of C/EBPβ or MYB in all cases was shown to counteract the anti-proliferative effects of the compounds, strongly suggesting that the inhibition of a common function of MYB and C/EBPβ underlies their activity in AML cells [66,83]. Overall, this argues strongly for an intimate and more general role of C/EBPβ, together with the co-activator p300, in supporting the oncogenic potential of MYB in AML, and leads to a model in which MYB, C/EBPβ and p300 cooperate in a transcriptional module to control the expression of genes that are critical for maintaining the viability of AML cells and preventing their differentiation. Interestingly, this view is supported by recent genome-wide chromatin-immunoprecipitation (ChIP) studies of AML cells which showed co-localization of MYB, C/EBPβ and p300 at many genomic loci in these cells [89].

These interesting and promising results give rise to the hope that the full potential of STLs as anti-leukemic lead compounds is yet to be explored, as they obviously act at very low concentrations by a novel mechanism of action, which currently gives a new boost to the research on STLs as anti-cancer agents. In vivo tests in leukemic animals will be an obvious next step in this direction.

Our findings on C/EBPβ inhibition by STLs are also very important with respect to the well-known anti-inflammatory activity of STLs. Helenalin and its derivatives such as HAC and various other STLs are quite well known for their anti-inflammatory activity (literature reviewed by [72,73,88]). It is quite important to note at this point that the inhibitory activity of STLs on another transcription factor, NF-B, has been known for more than 20 years and is widely associated with these anti-inflammatory properties of STLs [90,91,92]. This widely accepted mechanism of anti-inflammatory action of STLs, adopted even in textbooks of phytopharmacology [93,94], may have to be reconsidered. It is well established that the transcription machinery associated with C/EBPβ, besides its functions in cell proliferation and differentiation, is also deeply involved in the inflammatory response. C/EBPβ, for which the term nuclear factor-IL6 has also been coined, is involved in interleukin signalling in pathways such as those induced by the TNF-α- and IL-17 pathways (see pathway maps at http://www.kegg.jp, accessed on 21 March 2022), as well as [95,96,97,98,99]). It is important to note that the potency of helenalin and helenalin acetate to effectively inhibit C/EBPβ was found to be at least 10 times higher than that for NF-κB (EC_50_ values were determined between 0.1 and 0.4 μM for C/EBPβ vs. 4 μM for NF-κB) [81]. Helenalin, HAC and further derivatives are the active constituents responsible for the use of anti-inflammatory herbal medicines containing Arnica flowers [93,100]. It can hence be concluded from the present results that C/EBPβ inhibition may be even more important for the anti-inflammatory action of these and other STL-containing medicinal plants than their effect on NF-κB.

## 5. Withanolides: Withaferin A

Withaferin A (WFA, Figure 3) is a steroid lactone and representative of a small subclass of plant steroids, the so-called Withanolides, that are isolated from various genera of the family Solanaceae. WFA was the first compound of this group to be isolated from *Withania somnifera* (Winter cherry), a plant used in Ayurvedic medicine. The chemistry and biological activity of Withanolides has been reviewed comprehensively [101,102]. WFA and various other Withanolides were reported to possess anti-inflammatory as well as anti-cancer activity against several types of tumours. A plethora of possible mechanisms of action and targets related to cancer have been described ([101,102] and literature cited there).

When WFA was found to inhibit fluorescence activity in our MYB reporter gene assay, it was among the most active compounds, displaying an IC_50_ value of 1.8 µM, and displayed significant selectivity, the desired activity being about nine times stronger than its inhibitory effect in the MTS assay (i.e., cytotoxicity) [75]. It was therefore chosen for more detailed investigations on its mechanism of action [103]. Even though its structure is not very similar to the STLs described in the previous section, WFA also turned out, like e.g., HAC, to be a rather selective inhibitor of C/EBPβ activity. It did not impair the function of C/EBPα and inhibited C/EBPβ much more strongly than MYB. WFA was found to be a very potent inhibitor of C/EBPβ, with an EC_50_ of about 0.1 µM, i.e., in a similar range as helenalin and HAC. Like the STLs, WFA most strongly inhibited the LAP* isoform but, in contrast to the investigated STLs, it also interfered with the shorter LAP isoform, though to a lesser extent. Because LAP lacks the first 21 amino acids in comparison with LAP*, this suggested that the N-terminal domain must be involved in the observed effect. It was proven experimentally by a series of conclusive experiments that WFA also interrupts the interaction between the N-terminal domain (amino acids 1–50) of C/EBPβ and the TAZ2 domain of p300, thus resembling HAC in mechanistic terms. Since WFA also contains potentially reactive Michael addition sites as well as an epoxy group as a further potentially reactive structure element, mass spectrometric measurements were conducted in order to investigate whether the inhibition is caused by covalent modification of Cys residues. Indeed, both cysteines at positions 11 and 33 in the N-terminal domain of LAP*, as well as one further Cys at position 167, were found to be alkylated after treatment with WFA. Thus, the dependence of the observed inhibition of MYB-C/EBPβ-p300-related transcription on the presence of reactive structure elements (s) and covalent modification of the protein target was confirmed unambiguously in the case of WFA. Although it has not been proven experimentally so far, the cyclohexenone moiety of WFA is most likely the reactive structure responsible for cysteine modification and C/EBPβ inhibition, since the α, β-unsaturated lactone structure should be much less reactive due to the methyl substitution on the β-carbon atom. Since thiol groups as soft nucleophiles are generally much more susceptible towards soft electrophiles such as enone structures than to epoxides [80], which represent rather hard nucleophiles, it is unlikely that the epoxide group of WFA is involved in the observed effects. Like the STLs discussed above, WFA induced differentiation in an AML cell line (HL60), and this was dampened by the forced expression of MYB. This is consistent with the notion that WFA, by inhibiting the activity of C/EBPβ, also blocks the function of a MYB-C/EBPβ-p300 transcriptional module in AML cells [103].

In conclusion, these findings reveal a very interesting new mechanism of action and extend various previous attempts [101,102] to explain the anti-cancer activity of WFA. These data are of particular interest with regard to previous reports on an anti-leukemic potential of WFA [104] or other withanolides.

## 6. Quinone Methide Triterpenes: Celastrol

Two closely related pentacyclic nor-triterpenes with a quinone methide partial structure, namely celastrol (CEL) and its methyl ester pristimerin (Figure 3), were included in the initial screening for natural products inhibiting MYB-related transcription. Such compounds are typical of plants of the family Celastraceae such as the thunder god vine (*Tripterygium wilfordii*) from which CEL was originally isolated. Various biological effects, including anti-inflammatory as well as anti-cancer activity, have been reported for CEL, and various targets have been described [105].

Both CEL and pristimerin were among the most active compounds of the set screened in the MYB reporter gene assay, with IC_50_ values of 0.73 and 0.85 µM, respectively, and the selectivity of this effect over cytotoxicity was about 8–9-fold [75]. In mechanistic studies with CEL investigating the potential role of cysteine alkylation, a MYB mutant obtained after site-directed mutagenesis of all seven Cys residues of the protein to Alanine turned out to be equally responsive to the action of the compound as the wild type transcription factor. It was thus ruled out that CEL acts on MYB by forming covalent bonds to any of its cysteines [106]. The possibility that CEL inhibits MYB related transcription indirectly was thus also considered in this case. Quite interestingly, it then turned out that the decreased transcriptional activity can be restored by overexpressing the coactivator p300, which is known to interact with the TAD of MYB via its KIX domain. It was then proven by an in vitro binding experiment measuring the protein-protein interaction with a bacterial autodisplay system that CEL indeed disrupts the MYB-KIX interaction. Molecular docking simulations then indicated that CEL binds to a hydrophobic cleft on the KIX domain, which is normally the binding site for a α-helical LXXLL motif that forms the core of the MYB transactivation domain, and thus prevents the MYB-KIX interaction [106].

Initiated by the discovery that C/EBP inhibitors can also lead to a decrease in MYB transcriptional activity in our reporter gene assay, CEL was later also tested for inhibitory activity on C/EBPβ [107], and it was found in a C/EBPβ specific luciferase assay that triterpene is indeed a strong inhibitor of this transcription factor. Overexpression of p300, again, was able to override the inhibitory activity of CEL, which, in combination with some further experiments, showed that the interaction with the coactivator was also impaired in this case. Disruption of the interaction between C/EBPβ and the TAZ2 domain of p300 by CEL was demonstrated by direct protein-protein-binding experiments. Surprisingly, CEL independently targets the interactions of both cooperating transcription factors with their common binding partner, the coactivator protein p300. Since the quinone methide structure of CEL is known to react with thiol groups [105], it was then investigated whether the inhibitory effect on the C/EBPβ-TAZ2 protein-protein interaction is due to covalent modification of cysteines. A Cys-free, but functional, mutant of C/EBPβ was still sensitive to inhibition by CEL, proving that alkylation of Cys on the transcription factor cannot be responsible for the inhibitory affect. However, it was found that the TAZ2 domain, which contains several zinc finger-forming cysteine residues besides two “un-liganded” (i.e., not zinc-coordinated) cysteines, that these cysteines (C1789 and C1790) might be targets for CEL. Indeed, replacement of these two cysteines by alanine led to an almost complete abrogation of the ability of CEL to interrupt the C/EBPβ-TAZ2 protein-protein interaction and to inhibit the activity of C/EBPβ. Mutants with only one of the two cysteines replaced by alanine were still sensitive to CEL, showing that reaction with both adjacent Cys residues by CEL is required for the observed inhibitory effects [107]. Although alkylation of these residues by CEL has not yet been demonstrated by mass spectrometry for technical reasons, the mutation analysis makes it highly likely that both unliganded cysteines in the TAZ2 domain are direct targets of covalent modification by CEL. A recent structural model of the C/EBP-TAZ2 interaction [108] suggests that both cysteine residues are positioned in the vicinity of the interacting C/EBPβ sequences, making it possible that alkylation of these residues with the bulky Celastrol might sterically hinder the C/EBPβ-TAZ2 interaction. CEL is thus a dual inhibitor of the MYB-C/EBPβ-p300 transcriptional module inhibiting the interactions of two distinct domains of p300 with both transcription factors by different mechanisms, namely non-covalent binding to the KIX domain and (presumably) covalent binding to the TAZ2 domain.

In a very promising development, the selective activity of CEL was also detected in the above-mentioned retrovirus-induced murine leukemia models as well as in experiments with leukemic blasts from human patients [106]. Most importantly, CEL was also demonstrated to have an antileukemic activity in vivo. The urvival time of mice developing leukemia in a retrovirus-induced model of an aggressive AML was very significantly prolonged by CEL (3 × 0.5 mg/kg, i.p. per week over 4 weeks). Furthermore, it was shown that the MYB target genes c-myc and c-kit were downregulated in these animals and that their bone marrow cells grew much less quickly than those of untreated leukemic animals [106].

Taken together, these results make CEL an extremely promising candidate for further development against AML and, possibly, against further tumors.

## 7. Naphthoquinones: Plumbagin

The two most active natural products in the initial screening for compounds with inhibitory activity on MYB-related transcription were the naphthoquinones (NQs) shikonin and plumbagin [75,109] (Figure 3), with IC_50_ values ≤ 0.5 µM. Both had previously been reported to possess interesting bioactivity, including anti-cancer potential (reviewed in [110]). Naphthoquinones occur as secondary metabolites in various plant families. Since quinones are easily reduced to semi- and hydroquinones, they are associated with redox processes and the formation of reactive oxygen species (ROS), which accounts for the various biological activities of quinones [110]. However, NQs with unsubstituted β-carbons in the quinoid system can also undergo Michael additions. It was very interesting to note that among a variety of quinones tested, only such representatives showed inhibitory activity in the MYB reporter gene assay that have at least one unsubstituted electrophilic carbon in the quinone ring and are capable of reacting in a Michael type addition (see Figure 6).

Mechanistic studies were carried out mainly with plumbagin (PLB) [109], which was proven to inhibit the activity of the transactivation domain of MYB. MST experiments with PLB and shikonin clearly showed that the NQs bind directly to the TAD.

The possibility that a mechanism related to the NQ’s ability to generate reactive oxygen species (ROS) was involved could be ruled out by adding the antioxidants Trolox and Vitamin C, which did not impair the activity of PLB in the MYB assay. Interestingly, PLB’s inhibitory effect on MYB activity could be abrogated completely by pre-treatment with a low molecular weight thiol, N-actetyl cysteine (NAC). Furthermore, the studied NQs inhibiting MYB activity were indeed proven to form covalent adducts with NAC while inactive NQs did not [109,111], so that it could be expected that MYB inhibition is caused by alkylation of cysteines on the protein’s TAD. However, a Cys-free mutant of MYB was generated and found to be equally sensitive to inhibition by PLB as the wild type transcription factor, so that alkylation of cysteines on MYB could be ruled out as the cause of inhibition. It was then demonstrated that PLB, in a similar manner as described above for CEL, disrupts the interaction of the TAD of MYB with the KIX domain of co-activator p300. The possibility that PLB alkylates Cys residues on the p300/KIX side was discussed, but not investigated experimentally.

Interestingly, it had been found by previous authors that PLB is also a direct inhibitor of the histone acetylase function of p300 [112]. However, this latter activity required about 50 times higher concentration of the NQ than its observed effect on MYB-related transcription, so its relevance was questioned.

PLB was finally demonstrated to exert biological effects on AML cells expected to occur upon MYB inhibition. Several known MYB target genes were down-regulated, and the induction of differentiation was observed. Importantly, PLB-suppressed colony formation of leukemia cells derived from murine retrovirally-induced AML models more potently than that of normal hematopoietic progenitor cells from the bone marrow of healthy mice. Furthermore, colony formation of leukemic blasts from several human AML patients was also significantly suppressed by PLB, while normal CD34-positive hematopoietic progenitor cells from healthy donors were not affected, thus confirming its interesting anti-leukemic potential and providing new mechanistic insight into the anti-cancer activity of PLB [109].

## 8. Conclusions

In spite of great advances in the development of protein-based anti-tumor drugs such as “biologicals”, immunotherapy and other novel strategies, the search for new “small molecule” chemical entities with the potential as leads to anticancer drugs is still a very active and promising field [70,113]. Likewise, the discovery of new targets and compounds with new mechanisms of action to inhibit tumor growth is a very important goal [113].

As summarized in this manuscript, several studies over the last decade have shown that various natural products are potent inhibitors of the transcriptional machinery consisting of MYB, C/EBPβ and their coactivator p300. With these proteins’ cooperative transcriptional activity being tightly involved with tumorigenesis and tumor progression, especially in leukemia, such compounds can be considered interesting leads for the development of novel antitumor therapies. It is quite noteworthy, that, despite the fact that all natural products that displayed strong inhibition of this transcriptional activity in our investigations are potentially reactive molecules, they were shown to address different target sites on the three proteins (Figure 7). Thus, STLs like HAC, GOY and AT as well as the simplified synthetic STL analogue HM and the steroid lactone WFA were demonstrated to bind to the TAD of C/EBPβ, thereby inhibiting its interaction with the TAZ2 domain of coactivator p300. Similarly, the quinone methide triterpene CEL was shown to inhibit the C/EBPβ-TAZ2 interaction, but, very interestingly, this compound also inhibits the interaction between p300 and MYB by binding to the KIX domain of the former. Finally, the naphthoquinone PLB also interferes with the MYB-KIX interaction, probably by binding to the TAD of MYB.

It is rather important to note that all natural products with strong inhibitory activity on the transcription module under study had previously shown potential anticancer activity. In all cases, other possible mechanisms had been described to explain these effects. It is clearly not our intention to maintain that a particular mechanism of action is “the mechanism” responsible for the known antitumor effects of the mentioned compounds and their congeners. Such an attempt would actually represent the quite outdated view that one drug should have one particular target. It is now almost generally accepted that, especially in case of antitumor and anti-infective drugs, it is rather favourable to have more than one mechanism of action [114], and the natural products described in the present work appear to represent excellent examples for such polypharmacology. It is rather noteworthy, however, that the concentrations needed to obtain the observed inhibitory effects as consequences of MYB-C/EBPβ-p300 inhibition are very low, in some cases well below those needed to evoke other mechanisms. Even more importantly, the overexpression of MYB in AML cells (in case of HAC und HM also overexpression of C/EBPβ) was found to significantly decrease the sensitivity of the cells towards all of the natural products under study, which clearly shows that inhibition of the MYB-p300-C/EBPβ module does indeed contribute an important part to these compounds’ antitumoral mechanism of action. It can thus be concluded that the new mechanisms of action described here certainly make significant contributions to these natural products’ promising antitumor potential.

## Figures and Tables

**Figure 1 molecules-27-02077-f001:**
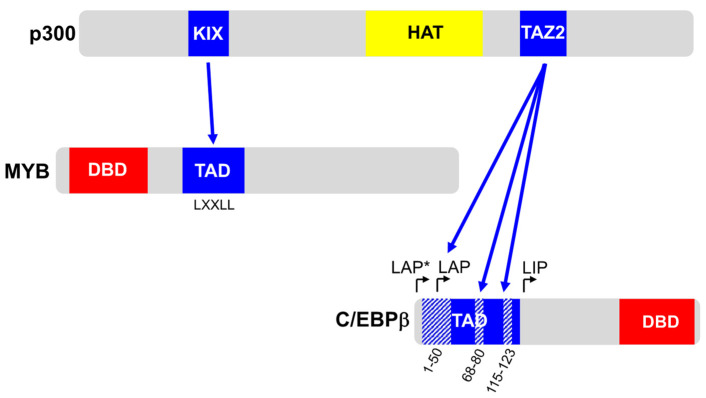
Protein-protein-interactions between MYB, C/EBPβ and coactivator p300. MYB binds via a conserved LXXLL motif of the transactivation domain (TAD) to the KIX domain of p300. The TAZ2 domain of p300 interacts with three parts of the C/EBPβ TAD located at amino acids 1–50, 68–80 and 115–123. DNA-binding domains (DBD) are shown in red. The histone acetyl-transferase domain (HAT) of p300 is shown in yellow.

**Figure 2 molecules-27-02077-f002:**
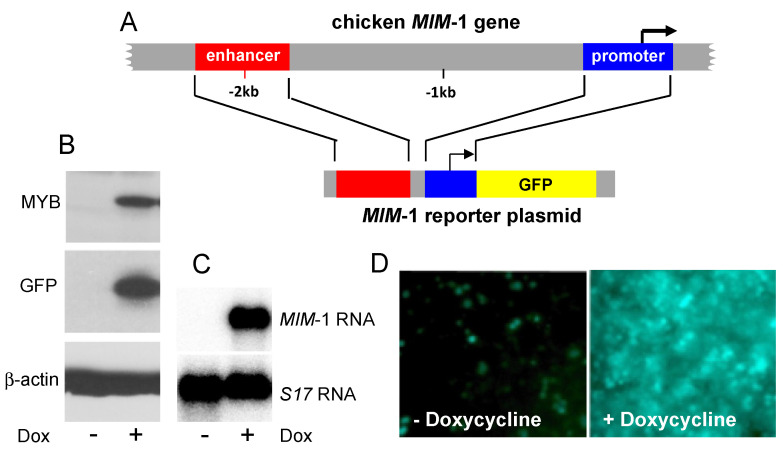
Fluorescence-based reporter system [68]. (**A**) Schematic structure of the *MIM*-1 gene promoter and enhancer and their fusion into the *MIM*-1 GFP-reporter plasmid. The reporter plasmid is stably integrated into the macrophage-like cell line HD11, which additionally expresses MYB in a doxycycline-dependent manner. (**B**) Western blot analysis showing MYB, GFP and β-actin expression in the absence or presence of doxycycline. (**C**) Northern blot analysis of endogenous *MIM*-1 and *S17* mRNA expression before and after treatment with doxycycline. (**D**) Microscopic pictures showing GFP fluorescence of control and doxycycline-treated cells.

**Figure 3 molecules-27-02077-f003:**
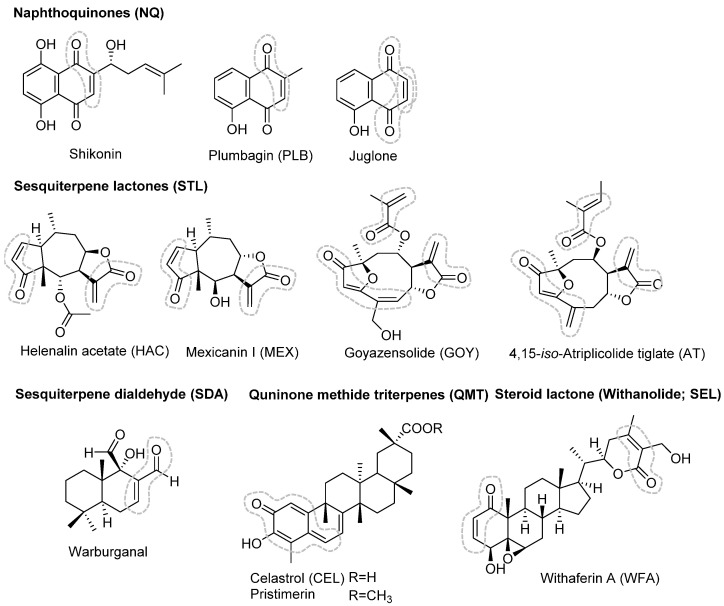
Chemical structures of the most active compounds (IC_50_ < 2 µM); compare Table 1. Enone systems (potential Michael acceptors) are marked with dashed lines in light grey.

**Figure 4 molecules-27-02077-f004:**
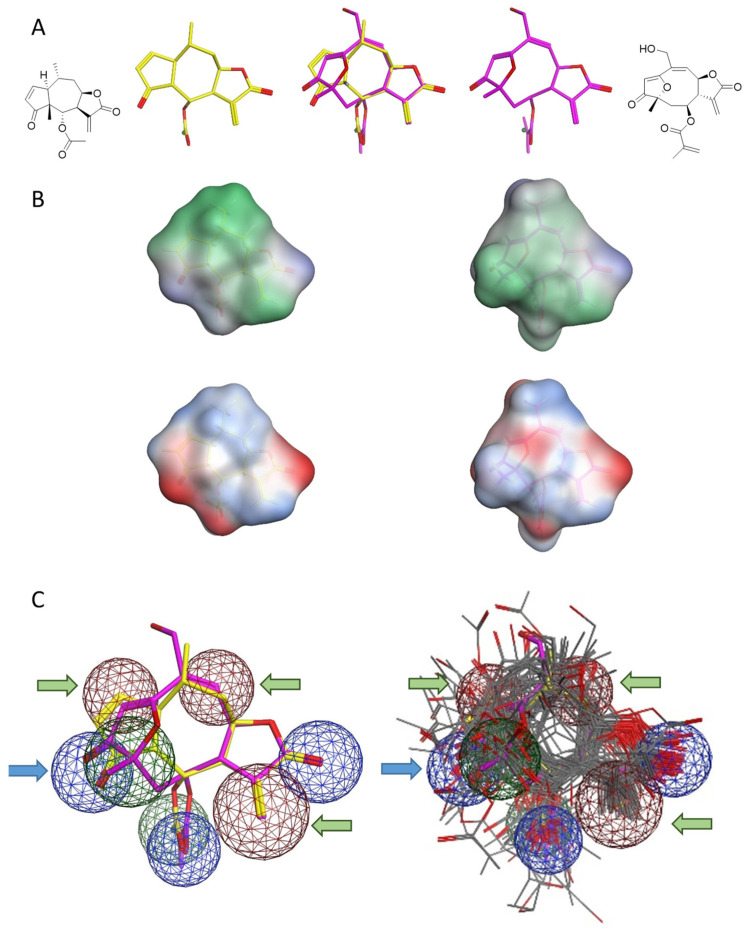
(**A**): Comparison of the chemical structures and 3D molecular models of HAC (**left**) and GOY (**right**), alignment of 3D structures is shown in the middle. (**B**): Comparison of the distribution of properties on the molecular surface of HAC (**left**) and GOY (**right**). In the upper row, the distribution of lipophilic (green) and hydrophilic surface area is shown, while the lower row shows electrostatic properties (red: negative, blue: positive) that are mapped on the molecular surface. (**C**): Joint pharmacophore model used in QSAR modeling. The pharmacophore spheres with relevant contributions to the QSAR model are marked by arrows: Reactive structure elements: Green arrows; H-bond acceptor: Blue arrow. **Left**: Structures of HAC and GOY underlying the pharmacophore model; **Right**: structures of all 60 STLs included in the study after alignment with those of HAC and GOY mapped into the pharmacophore model. Modified, according to [77].

**Figure 5 molecules-27-02077-f005:**
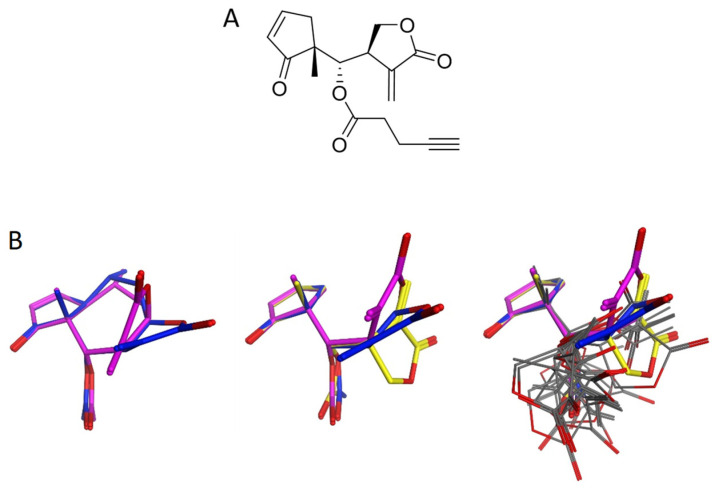
(**A**) Structure of a synthetic helenalin analogue (“helenalin mimic”, HM). (**B**) Comparison of the two lowest energy conformers of HAC (**left panel**) and the acetate analog of HM (**middle**, **right**). The middle panel shows the lowest energy conformer (yellow) and the two HAC-like conformers, which are about 3.5 and 4.5 kcal/mol (blue, pink, respectively) higher in energy. In the right panel, all conformers of HM within an energy window from the global minimum are superposed. (All structural models shown in (**B**) were aligned by superposition of the atoms of the cyclopentenone ring).

**Figure 6 molecules-27-02077-f006:**
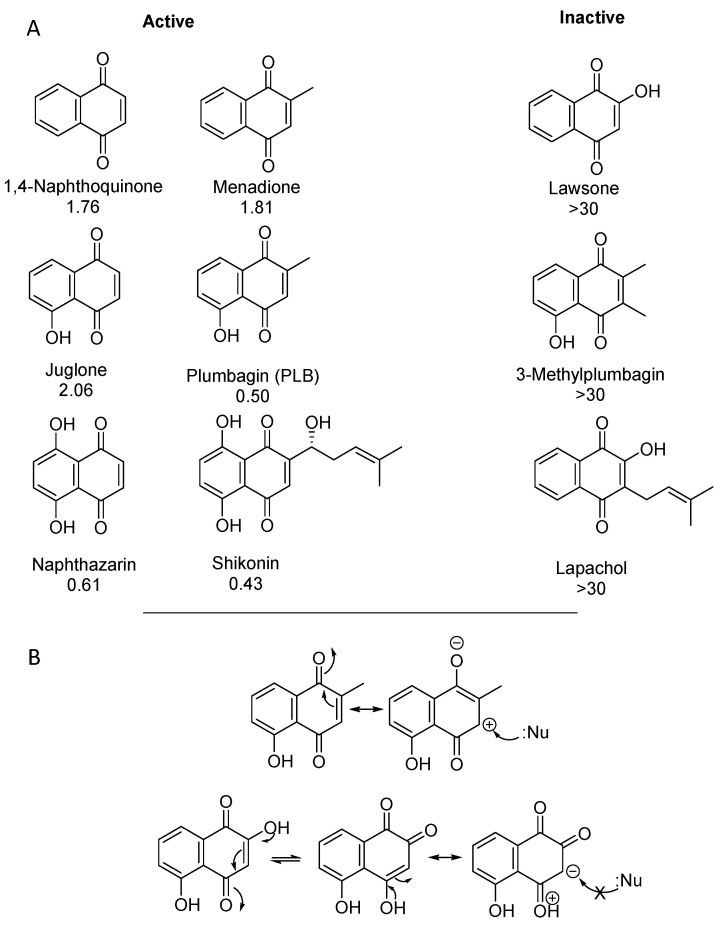
(**A**) Chemical structures and inhibitory activity (EC_50_ values in µM) of naphthoquinones in the MYB reporter gene assay. (**B**) **Top**: Plumbagin as example for active NQs has an unsubstituted β-position relative to the upper keto group that can react with a nucleophile (Nu). **Bottom**: Lawsone also has an unsubstituted β-position relative to the upper keto group, but the neighboring hydroxyl group is in conjugation with the second keto group, which increases electron density at the β-carbon which is no longer reactive towards Nu.

**Figure 7 molecules-27-02077-f007:**
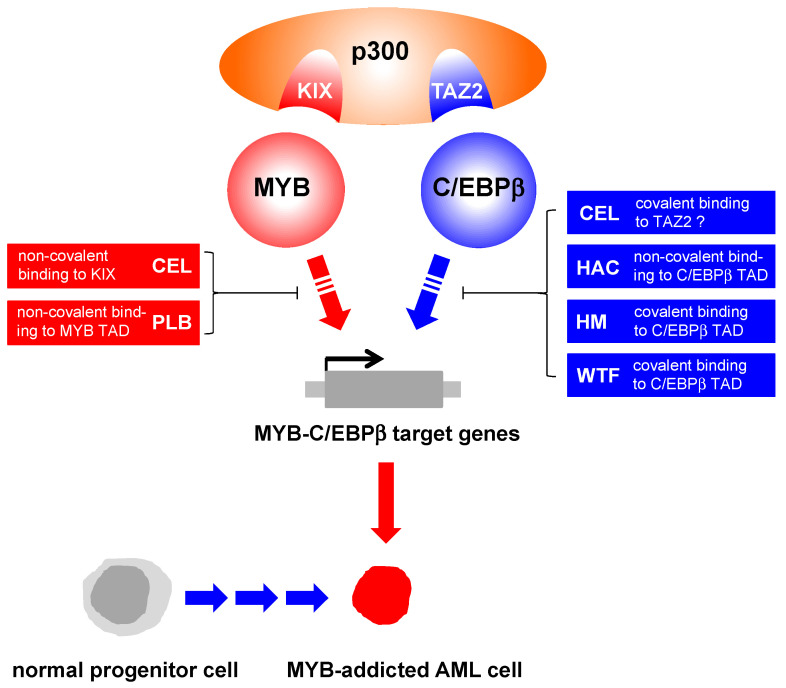
Interference of the various natural product inhibitors with the cooperation of MYB, C/EBPβ and the coactivator p300.

**Table 1 molecules-27-02077-t001:** Active compounds with IC_50_ < 5 µM from a screening of 100 natural products against MYB related transcriptional activity. For comparison, impairment of cell viability in an MTS assay is reported as well. Data are IC_50_ values in µM, means from three independent determinations ± standard deviation (sd) in both cases. SI is the selectivity index, IC_50_ (MTS)/IC_50_ (MYB). Data according to [75].

Compound	NP Class ^a^	MYB	±SD	MTS	±SD	SI
Shikonin	NQ	0.30	±0.06	16.11	±8.32	53.7
Plumbagin	NQ	0.56	±0.18	11.45	±5.30	20.4
Goyazensolide	STL	0.63	±0.19	10.05	±3.96	16.0
Helenalin acetate	STL	0.71	±0.03	7.34	±0.95	10.3
Pristimerin	QMT	0.73	±0.15	6.24	±2.53	8.5
Celastrol	QMT	0.85	±0.10	6.62	±2.35	7.8
Juglone	NQ	0.98	±0.03	>30		>30
Warburganal	SDA	1.54	±0.97	12.80	±5.27	8.3
4,15-iso-Atriplicolide tiglate	STL	1.57	±0.47	12.07	±6.46	7.7
Withaferin A	SEL	1.77	±0.16	15.67	±11.09	8.9
Mexicanin I	STL	1.80	±0.06	21.13	±2.19	11.7
4,15-iso-Atriplicolide methacrylate	STL	2.21	±0.32	10.38	±3.96	4.7
Helenalin	STL	2.37	±0.39	21.11	±2.63	8.9
Budlein A	STL	2.39	±0.25	9.00	±2.96	3.8
Helenalin isobutyrate	STL	2.54	±0.28	>30		>12
Primin	BQ	2.73	±0.21	22.11	±4.89	8.1
Tagitinin C	STL	2.92	±1.08	30.91	±10.93	10.6
Enhydrin	STL	2.95	±1.23	7.49	±0.29	2.5
Uvedalin	STL	3.83	±1.22	11.42	±4.19	3.0
Perezone	BQ	4.57	±0.29	30.0	±13.81	6.6
Melcanthin C	STL	4.68	±1.63	24.80	±4.36	5.3
Nobilin	STL	4.73	±1.40	>30		>6

^a^ NQ: Naphthoquinone; STL: Sesquiterpene lactone; QMT: Quinone methide triterpene; SDA: Sesquiterpene dialdehyde; SEL: Steroid lactone; BQ: Benzoquinone.

## Data Availability

Not applicable.

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
