# Peer review of "Natural Products with Antitumor Potential Targeting the MYB-C/EBPβ-p300 Transcription Module"

_molecules, 2022, doi:10.3390/molecules27072077_

Round 1
Reviewer 1 Report
This is a well-written and thorough review about the MYB oncoprotein and transcription factor, its interactions with p300/CBP and the inhibitors that have been identified to inhibit MYB activity. The authors have done an excellent job of summarizing the field and showing how screens can be developed to identify inhibitors of transcription factors like MYB.
Author Response
Reviewer 1
This is a well-written and thorough review about the MYB oncoprotein and transcription factor, its interactions with p300/CBP and the inhibitors that have been identified to inhibit MYB activity. The authors have done an excellent job of summarizing the field and showing how screens can be developed to identify inhibitors of transcription factors like MYB.
We thank the reviewer for the very positive assessment.
Reviewer 2 Report
This paper reviews Natural Products with Antitumor Potential targeting the MYB-C/EBPbeta-p300 Transcription Module. In the first part, the authors appropriately situate the reader in the importance of the MYB-C/EBPbeta-p300 Transcription Module in cancer.
Subsequently, they discuss the importance of sesquiterpene lactones as inhibitors of this transcription module and their possible mechanisms. Subsequently, they discuss other natural products such as MYB-C/EBPbeta-p 300 transcription assembly inhibitors. They still discuss the possibility that not only the transcription factor is the only target, mainly of STL. Although the work is well written, much of the information was previously presented by the same authors. Perhaps, something is expected in a review. However, there is nothing new in this review. A new approach would have been expected, or possibly present the information to obtain new directions to investigate. It is also noteworthy that they do not mention, in a prominent way, in vivo tests. It is well known that the presence of Michael acceptor systems such alpha, beta ketone moiety or alpha, beta-unsaturated lactone moiety are targets of nucleophilic attack by cysteine ​​residues or other nucleophilic groups present in enzymes. The alpha, beta-unsaturated lactone moiety group is present in the active STL; however, it is not selective, and it is known that it can attack various enzymes, thus explaining its high toxicity. Thus, it is not surprising that it is now being tested on the MYB-C/EBPbeta-p300 transcription module and possibly on other enzymes such as p 300. In summary, this article is well written and easy to read, but unfortunately, it is little novelty that can contribute.
Author Response
Reviewer 2
This paper reviews Natural Products with Antitumor Potential targeting the MYB-C/EBPbeta-p300 Transcription Module. In the first part, the authors appropriately situate the reader in the importance of the MYB-C/EBPbeta-p300 Transcription Module in cancer.
Subsequently, they discuss the importance of sesquiterpene lactones as inhibitors of this transcription module and their possible mechanisms. Subsequently, they discuss other natural products such as MYB-C/EBPbeta-p 300 transcription assembly inhibitors. They still discuss the possibility that not only the transcription factor is the only target, mainly of STL. Although the work is well written, much of the information was previously presented by the same authors. Perhaps, something is expected in a review. However, there is nothing new in this review.
Indeed, this is the nature of a review article which serves primarily to summarize current knowledge on a particular topic.
A new approach would have been expected, or possibly present the information to obtain new directions to investigate. It is also noteworthy that they do not mention, in a prominent way, in vivo tests.
In vivo tests are mentioned wherever they were possible and the necessity to perform such tests in cases where they were not yet possible is also mentioned.
It is well known that the presence of Michael acceptor systems such alpha, beta ketone moiety or alpha, beta-unsaturated lactone moiety are targets of nucleophilic attack by cysteine ​​residues or other nucleophilic groups present in enzymes. The alpha, beta-unsaturated lactone moiety group is present in the active STL; however, it is not selective, and it is known that it can attack various enzymes, thus explaining its high toxicity.
The concentrations of STLs needed to obtain the described effects are much lower than those needed for various other effects observed previously which already indicates that there is indeed selectivity in the described effects. Furthermore, specific tumour selectivity has been shown in all cases (not only for STLs but for all types of natural products under study) by comparison of the effects on leukemic blasts with healthy bone marrow cells. Most importantly, in case of all types of natural products described here, that overexpression of MYB in AML cells (in case of HAC und HM also overexpression of C/EBPb) significantly lowers the sensitivity of the cells towards the respective compounds. This clearly indicates that inhibition of the MYB-p300-C/EBPb module does indeed contribute a significant aspect to these compounds’ antitumoral mechanism of action. A short statement on this latter point has been added to the last section (changes tracked) and hope this aspect is now clearer.
Thus, it is not surprising that it is now being tested on the MYB-C/EBPbeta-p300 transcription module and possibly on other enzymes such as p 300.
Maybe it is not surprising for the reviewer, but it was quite a surprise to discover, for the first time, low molecular weight inhibitors for this interesting target.
In summary, this article is well written and easy to read, but unfortunately, it is little novelty that can contribute.
We thank the reviewer for the time and effort.